# Teaching People LLM's Errors and Getting it Right

## Abstract

People use large language models (LLMs) when they should not. This is partly because they see LLMs compose poems and answer intricate questions, so they understandably, but incorrectly, assume LLMs won't stumble on basic tasks like simple arithmetic. Prior work has tried to address this by clustering instance embeddings into regions where an LLM is likely to fail and automatically describing patterns in these regions. The found failure patterns are taught to users to mitigate their overreliance. Yet, this approach has not fully succeeded. In this analysis paper, we aim to understand why.

We first examine whether the negative result stems from the absence of failure patterns. We group instances in two datasets by their meta-labels and evaluate an LLM's predictions on these groups. We then define criteria to flag groups that are sizable and where the LLM is error-prone, and find meta-label groups that meet these criteria. Their meta-labels are the LLM's failure patterns that could be taught to users, so they do exist. We next test whether prompting and embedding-based approaches can surface these known failures. Without this, users cannot be taught about them to reduce their overreliance. We find mixed results across methods, which could explain the negative result. Finally, we revisit the final metric that measures teaching effectiveness. We propose to assess a user's ability to effectively use the given failure patterns to anticipate when an LLM is error-prone. A user study shows a positive effect from teaching with this metric, unlike the human-AI team accuracy. Our findings show that teaching failure patterns could be a viable approach to mitigating overreliance, but success depends on better automated failure-discovery methods and using metrics like ours.

## 1 Introduction

People are using LLMs to make decisions, ranging from well-defined tasks in specialized domains such as law and medicine to ad-hoc queries (Ayers et al., 2023; Siino et al., 2025; Zhao et al., 2024). Despite this increased interaction with LLMs, users still lack a robust understanding of their capabilities, and consequently, they rely on them inappropriately (Vaccaro et al., 2024; Si et al., 2024; Kim et al., 2024). Real-world incidents show that such reliance can cause misinformation and reputational harm (Le Jeune et al., 2025). This calls for methods to support more accurate mental models of LLM capabilities. One proposed approach to address this is *the AI-integration teaching pipeline* (Mozannar et al., 2023), overviewed in Fig. 1 and §2. After partitioning the data, this pipeline aims to *automatically* identify patterns in the instances where an LLM is prone to fail and then teach users to avoid using the LLM for instances exhibiting these patterns. If these failure patterns are accurate and users receive effective guidance, users could better understand what not to use the LLM for.

However, Mozannar et al. (2023)'s attempt to teach users in this way has not improved the accuracy of people when assisted by an LLM. Specifically, they report that after teaching, human accuracy in classifying certain images improves when participants receive assistance from the Faster R-CNN model (Ren et al., 2015), but not when they are assisted with an LLM to answer MMLU questions (Hendrycks et al., 2021). This negative result casts doubt on the AI-integration teaching as a general strategy for mitigating overreliance and improving the accuracy of human-AI teams. We aim to explain where it fails and identify the changes needed to make it a viable approach.

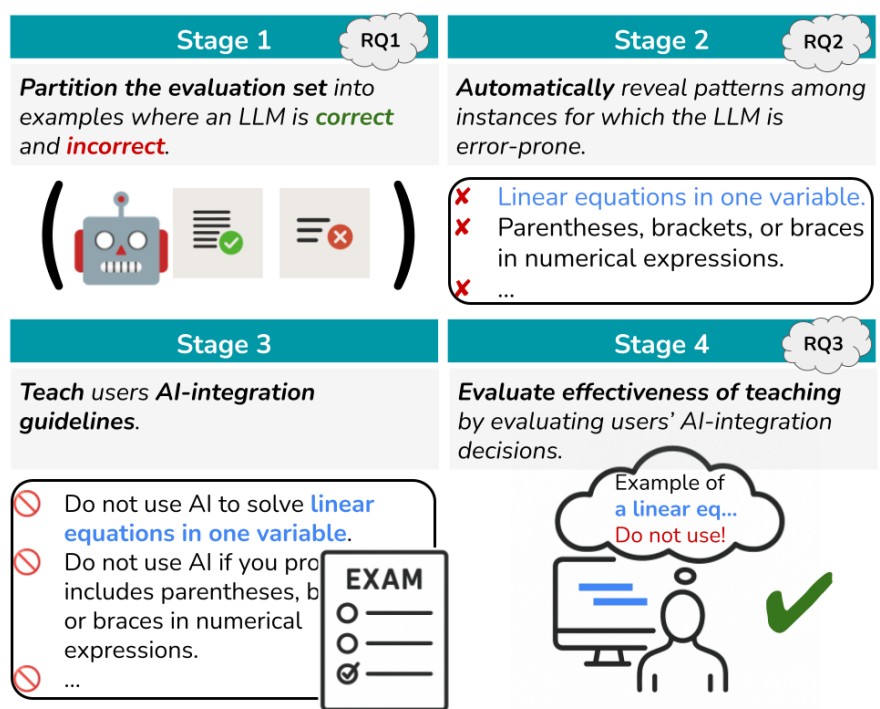

Figure 1: **The AI-integration teaching pipeline**. We investigate the underlying causes of its reported limited benefits in reducing overreliance on LLMs.

To this end, we consider each stage of the pipeline and examine the factors that must be accounted for when interpreting the results of its downstream effectiveness. Specifically, we ask:

- **RQ1: (Stage 1)** Are there patterns that describe sizable groups of evaluation instances where an LLM has elevated error rates?[1] If LLM errors are scattered across many types, each making up only a small fraction of the total error, they are likely too numerous to remember and too specific to be widely applicable. Basing the evaluation of the AI-integration teaching in this case is unlikely to yield benefits and could explain why it was reported ineffective by Mozannar et al. (2023).
- **RQ2: (Stage 2)** Once we know these failure patterns exist and have identified some of them, can we automatically and accurately generate them? If not, generated patterns are noisy, so teaching users about them is unlikely to be beneficial. This could also explain the negative result.
- **RQ3: (Stage 4)** Do we get different insights in teaching effectiveness by assessing whether people understand when to use or avoid an LLM instead of task accuracy? One might correctly judge when to avoid using an LLM, but if they cannot solve the problem themself, LLM-assisted human accuracy will not reflect whether they recognized they should not use the LLM.

To answer RQ1, we first define measurable instance group characteristics: *coverage* and *error ratio* (§3). Teaching a pattern characterizing a group with a "high-enough" coverage and error ratio could reduce users' overreliance. We investigate whether such groups exist in MMLU and MathCAMPS (Mishra et al., 2024). We group their instances by available human-annotated meta-labels and use predictions of various LLMs on these meta-label groups. We find meta-label groups that have a "high-enough" coverage and error ratio (§4). Their meta-labels are the associated LLM's failure patterns that could be taught to users, so such patterns exist for explored LLMs and these datasets. Therefore, *the failure of the AI-integration teaching pipeline in prior work is not due to an absence of failure patterns (Stage 1) for these datasets and LLMs.*

Next, we address RQ2 in §5 by testing how well two prompting and two embedding-based methods identify and generate these known failure patterns. A prompting method comes close to accurately retrieving the known failure patterns, but Mozannar et al. (2023)'s embedding-based method does not. This suggests *Stage 2 of the AI-integration teaching pipeline may be responsible for its failure in prior work.*

---

[1]"Sizable" and "elevated" are used informally here, but we specify them in §3. Similarly, "high-enough" used later.

Lastly, we investigate the appropriateness of LLM-assisted human accuracy for measuring the success of the AI-integration teaching pipeline through a user study that directly compares it to an alternative metric (§6). Specifically, we measure how accurately a user can anticipate whether an LLM will succeed or fail on a given instance. We find through a user study that this measurement highlights the effectiveness of the AI-integration teaching for a task where human-AI team accuracy previously did not. That is, *the choice of final measurement in Stage 4 alone can lead to different interpretations of the success of the AI-integration teaching pipeline.*

Our findings show that the AI-integration teaching pipeline could be the urgently needed approach for reducing overreliance on LLMs. Its success, however, depends on more accurate automated discovery of failure patterns and on the adoption of more suitable metrics for evaluating teaching effectiveness, such as the one we propose.

## 2 Background and Terminology

This section introduces the AI-integration teaching pipeline and clarifies our use of terminology for concepts that are difficult to formalize.

**The AI-Integration Teaching Pipeline.** To examine why teaching users the patterns common to instances where an LLM is error-prone might fail to mitigate overreliance, we first formalize the general steps involved as a four-stage pipeline shown in Fig. 1. First, a dataset-model pair is selected, and the dataset is partitioned according to model correctness. This stage determines which examples are used to discover and describe failure patterns of the LLM. Next, an automated method generates natural language descriptions of these failure patterns. Automation of this process is essential for quickly and scalably identifying new failure patterns which can result from changing model capabilities and emergence of new tasks. In Stage 3, these descriptions are turned into *AI-integration guidelines*. For example, if a pattern is described as "linear equations in one variable", the guideline becomes "do not use the LLM for linear equations in one variable". Users are then taught these guidelines and tested on a set of practice questions. This process aims to help users better recognize when to trust or ignore model predictions.[2] Finally, a user study evaluates the effectiveness of Stage 3, and thereby the entire pipeline, according to a chosen metric. While previous work has addressed parts of the pipeline (see §7), Mozannar et al. (2023) are the first to instantiate it in its entirety. In the LLM setting, their full evaluation is limited to `GPT-3.5-turbo` on MMLU, where their pipeline is ineffective.

**Terminology.** In Stage 2, an automatic method produces natural-language descriptions, each characterizing the properties common to a group of instances that an LLM "often" handles incorrectly. We say that the LLM is *error-prone* on such a group and call the resulting descriptions *failure patterns*. The threshold for how "often" the LLM must fail to consider it error-prone is context-dependent and is discussed further in §3. However, we stress here that "often" does not necessarily mean "always", so some instances within a group where the LLM is error-prone may still be handled correctly.

Given a group of instances, we want to determine whether a pattern among its instances is *worth teaching users about* to mitigate overreliance. In §3, we propose "high-enough" coverage and error-ratio as criteria for identifying such groups. We call a pattern that is derived from a group that meets these criteria a *worth-to-teach* failure pattern.

Finally, the pipeline mitigates overreliance on the LLM by teaching users when it is error-prone, assuming the worst case where they would otherwise use it. For brevity, we say "mitigate overreliance" instead of "mitigate *potential* overreliance".

## 3 Characteristics of Groups Worth Generating Failure Patterns For

In this section, we define quantitative criteria for identifying which instance groups are worth generating a failure pattern for. Intuitively, the most useful patterns to teach users are those that (i) occur frequently in the data and (ii) describe cases where the model's error rate is elevated. The former can be quantified by

---

[2]This process is sometimes called *onboarding* (Cabrera et al., 2023; Cai et al., 2019; Mozannar et al., 2022).

the proportion of total errors that fall under a given failure pattern, while the latter can be captured by the ratio of incorrect to correct instances matching that failure pattern. We refer to these two characteristics as *coverage* and the *error-ratio*, respectively.

Formally, given an evaluation dataset $D$, let $D_c$ and $D_w$ be the subsets of $D$ where the model is wrong and correct, respectively. Let $p_i$ denote a data pattern, which is a candidate failure pattern that could be useful to teach to users. Let $G_{p_i}$ be a set of all instances in $D$ that have this pattern, i.e., $G_{p_i} = \{x \in D \mid \texttt{has\_pattern}(p_i, x) = \texttt{true}\}$.

For each group $G_{p_i}$, we define:

$$\texttt{coverage}(G_{p_i}) = \frac{|G_{p_i} \cap D_w|}{|D_w|}, \tag{1}$$

$$\texttt{error-ratio}(G_{p_i}) = \frac{|G_{p_i} \cap D_w|}{|G_{p_i} \cap D_c|}. \tag{2}$$

Coverage measures the proportion of a model's total errors in $D$ that the pattern $p_i$ describes. Teaching users to avoid the model for instances of a pattern tied to a high-coverage group is more promising for reducing overreliance, because its cases account for a larger share of the model's total errors. However, if coverage is too high, it may signal that the pattern is defined too broadly to be actionable. For instance, a group described as "complex problem solving" might cover many errors, but also include many correct instances.

Therefore, we also measure the error ratio: the proportion of incorrect to correct predictions in $G_{p_i}$. It measures how often the model fails on instances with the pattern $p_i$ and how well $p_i$ distinguishes errors from correct cases. Patterns tied to high-error-ratio groups are preferable as they show where the model fails frequently enough that users should be warned. As the error ratio decreases, the associated pattern becomes less compelling to teach — if the model usually gets these instances right, the risk to users is low. Still, the threshold for what constitutes a "high-enough" error ratio depends on context. An error ratio of 0.33 (1 out of 4 examples with that pattern is wrong) might seem too low to warrant teaching users about, but in high-stakes domains like medical diagnosis, even this level of risk may warrant user attention. Low error ratios may also be needed to identify the rare failure modes of otherwise highly accurate models.

From the above discussion of coverage and error ratio, two factors emerge that must be balanced when selecting patterns to teach in Stage 3. First, the coverage and error ratio should be "high enough". Second, what counts as "high enough" depends on the application. We therefore recommend either (i) setting thresholds in consultation with stakeholders or (ii) reporting the effectiveness of teaching how to integrate AI across a range of minimum thresholds. The latter approach is more exhaustive but could be prohibitively expensive, as it requires evaluating the downstream impact of teaching failure patterns at each threshold.

## 4 Existence of Groups Worth Generating Failure Patterns For

The AI-integration teaching pipeline aims to discover a model's failure patterns and teach them to users to reduce their overreliance. However, this assumes that the model's predictions form groups whose patterns are worth generating. Yet, such groups may not exist, so the pipeline's later stages cannot yield positive outcomes. In this section, we use coverage and error-ratio (§3) to assess whether such groups exist for various LLM and two datasets with annotated meta-labels.

Specifically, meta-labels may correspond to patterns that describe sizable groups of instances (high coverage) where an LLM is error-prone (high error ratio). To check, we first group according to the chosen meta-labels and then compute the error ratio and coverage based on the groupings to see if they satisfy our particular threshold. If so, such a model-dataset combination is suitable for studying the effects of teaching, and these meta-labels can be used as ground-truth failure patterns. If not, this only means the meta-labels do not align with failure patterns, and instance groups that are worth generating failure patterns for may still exist.

**Setup.** Following the above approach, we investigate candidates for failure patterns in two datasets relative to various LLMs. These datasets are:

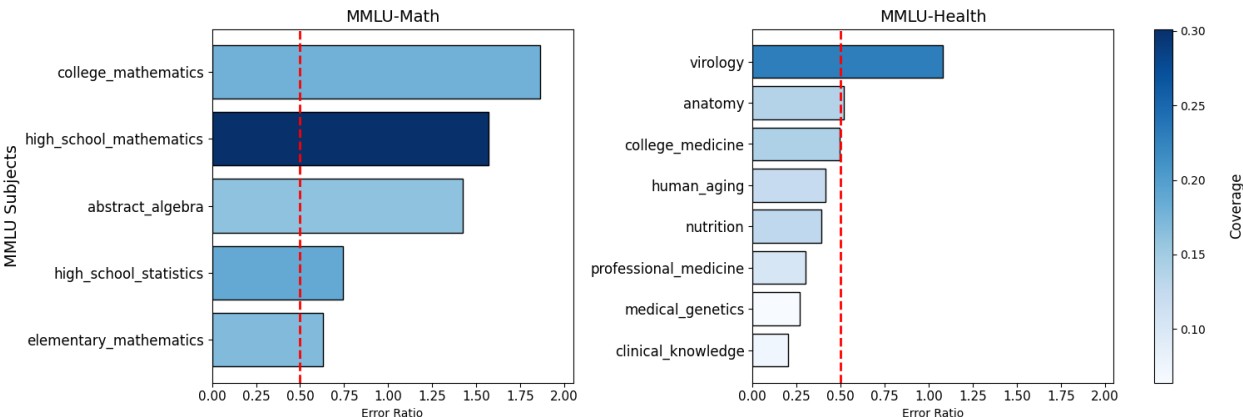

Figure 2: Subjects in **MMLU** Math (left) and Health (right) sorted by error ratio for `GPT-3.5-turbo`.

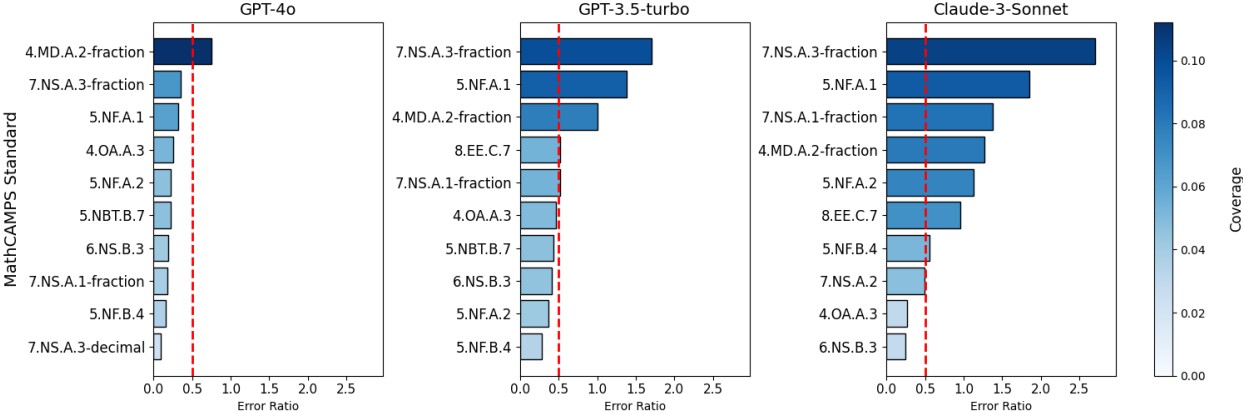

Figure 3: Top-10 **MathCAMPS** standards by error ratio for `GPT-4o`, `GPT-3.5-turbo`, and `Claude-3-Sonnet`.

- **MMLU** (Hendrycks et al., 2021). A popular multiple-choice QA benchmark used to test LLM capabilities across many domains. We focus on two representative sub-categories, Mathematics and Health, and use the "subject" meta-labels for each instance to see if some subjects are sought failure patterns. Our choice to use MMLU follows prior work (Mozannar et al., 2023), whose results we aim to understand better.
- **MathCAMPS** (Mishra et al., 2024). This dataset contains math problems that are mapped to 49 educational standards for grades K-8. The standards describe specific skills and concepts that are needed to solve a particular problem, offering a rich set of meta-labels which could align with failure patterns of LLMs. These meta-labels offer a meaningfully different proxy for error patterns compared to MMLU. We use the entire dataset of 4900 examples. See Table 3 for example standards.

Table 4 (Appendix) shows an example per dataset. We use LLM predictions released by Mishra et al. (2024) on MathCAMPS and Mozannar et al. (2023) on MMLU. We treat meta-label groups with an error ratio of 0.5 or higher as suitable to teach users. Here, 0.5 means an LLM is wrong at least once in every three attempts for that instance type.

**Results.** We show meta-label groups sorted by error ratio in Figures 2–3.[3] On the MMLU-math subset, all meta-label groups have an error ratio higher than 0.5 for `GPT-3.5-turbo`, collectively covering 100% of total error. The health subset has two such groups, covering 36.7% of total error. One such meta-label group exists for `GPT-4o` in MathCAMPS, accounting for 11.2% of total error, whereas five such groups exist for `GPT-3.5-turbo`, accounting for 37.6% of total error. The existence of such groups makes generating

---

[3]App. G shows results of other MMLU sub-categories. We exclude Standard 8.EE.C.8 (0% accuracy) to avoid skewing the analysis with a large outlier in terms of error ratio.

failure patterns for these dataset-LLM combinations worthwhile and these setups suitable for studying the AI-integration teaching pipeline.

However, the difference in the number of such groups across LLMs for the same dataset hints that model choice matters, even with the same dataset and error-ratio threshold. To illustrate why, consider two researchers, R1 and R2, both evaluating their AI-integration teaching pipelines on MathCAMPS using a 0.5 error-ratio cutoff. R1 uses `GPT-4o`, while R2 uses `GPT-3.5-turbo`. This is a realistic divergence in this emerging area, where researchers often default to the best available model. Despite using the same setup, R1 and R2 could reach different conclusions about teaching effectiveness simply due to differences in the number or type of mistakes models make. Foremost, R1's participants only need to learn one failure pattern to meaningfully reduce overreliance, whereas R2's need five, which imposes a greater cognitive burden on users. Moreover, one might expect that discovering and describing a single prominent failure pattern may be easier than five, a hypothesis we investigate (and disprove) in the next section.

## 5   Generating Failure Patterns

In §4, we find that meta-label groups with "high-enough" coverage and error ratio exist for various LLM-dataset pairs. We now address whether these known failure patterns can be identified automatically. Specifically, we evaluate prompting-based methods, where an LLM is prompted to describe failure patterns from given instances, and embedding-based, where embeddings are clustered into groups and an LLM is used to describe the patterns in each group. We also study providing methods with the LLM's chains of thought (CoT; Wei et al., 2022) alongside dataset examples. We assess these methods on MathCAMPS using five LLMs and on two MMLU subsets using one LLM.

### 5.1   Prompting-based Methods

The prompts used are provided in Appendix A and additional method details in Appendix D.

**Direct.** We give all mishandled instances, $D_w$, to `gpt4o-2024-08-06` (OpenAI, 2024), the state-of-the-art LLM at the time, and prompt it to generate descriptions of their shared properties.

**D5.** We adapt Zhong et al. (2023)'s method for describing the difference between text corpora to the task of generating failure patterns by setting the text groups to be contrasted as instances where the model was wrong ($D_w$) and correct ($D_c$). To facilitate comparison, we adjust their method to use `gpt-4o-2024-08-06` to generate the patterns.

### 5.2   Embedding-based Methods

**IntegrAI.** Mozannar et al. (2023) introduce an algorithm which clusters instances into regions where performance of humans and AI models differs. A summary description for each region is produced using an LLM and is iteratively refined by contrasting the region description with counterexamples — instances similar to the description but outside the cluster. In order to restrict this method to discover failure patterns as opposed to patterns of instance groups where models were better than humans, we use this method in the single-model setting with the `all_0` prior.

**Mapper Graphs.** Mapper graphs (Singh et al., 2007) are a popular tool from topological data analysis that have been used to summarize the underlying structure of an embedding space by leveraging its topology (Rathore et al., 2021; 2023; Zhou et al., 2023; Purvine et al., 2023; Yan et al., 2025a). We construct a mapper graph from a set of embeddings originating from a dataset. Intuitively speaking, a node in a mapper graph corresponds to a cluster (i.e., topological neighborhood) of the embedding space, and an edge indicates the overlap between two neighborhoods. Two embeddings belong to the same mapper node (cluster) if (i) they are close to each other in terms of the underlying metric, and (ii) their respective $L_2$-norm fall in the same range (i.e., the model exhibits an equally strong response to each of their respective data points). For a given evaluation dataset, $D$, we first construct a mapper graph and identify *erroneous mapper nodes* that contain wrong instances. To balance coverage and error ratio, we merge erroneous nodes iteratively using a greedy strategy that maximizes their error-score, defined as $\text{error-score}(G_{p_i}) = \text{error-ratio}(G_{p_i}) \cdot \text{coverage}(G_{p_i})$.

| | | MathCAMPS | | | | | MMLU + GPT-3.5-Turbo | |
|---|---|---|---|---|---|---|---|---|
| | | *GPT-4o (1)* | *GPT-3.5-Turbo (5)* | *Sonnet (7)* | *Opus (4)* | *Haiku (7)* | *Math (5)* | *Health (2)* |
| **Direct** | Base | 3.00 ± 0.00 | **3.67 ± 0.12** | 2.86 ± 0.00 | **3.92 ± 0.14** | 3.66 ± 0.08 | **4.47 ± 0.12** | **3.50 ± 0.00** |
| | ↳ CoT | **3.67 ± 0.58** | 3.13 ± 0.23 | 3.48 ± 0.08 | 3.00 ± 0.25 | **3.86 ± 0.15** | 4.40 ± 0.00 | 2.17 ± 0.29 |
| **D5** | Base | 2.67 ± 0.58 | 2.40 ± 0.20 | 3.29 ± 0.00 | 3.33 ± 0.14 | 3.43 ± 0.00 | 2.93 ± 0.12 | 2.83 ± 1.04 |
| | ↳ CoT | 1.33 ± 0.58 | 2.80 ± 0.20 | 2.81 ± 0.09 | 2.42 ± 0.14 | 3.00 ± 0.00 | 2.93 ± 0.12 | 1.50 ± 0.00 |
| **IntegrAI** | Base | 3.00 ± 0.00 | 3.13 ± 0.12 | 3.00 ± 0.14 | 3.00 ± 0.00 | 3.05 ± 0.08 | 2.40 ± 0.20 | 2.67 ± 0.29 |
| | ↳ CoT | 3.00 ± 0.00 | 2.80 ± 0.20 | 2.71 ± 0.15 | 3.25 ± 0.00 | 2.91 ± 0.08 | 3.07 ± 0.12 | 2.83 ± 0.29 |
| **Mapper** **Graphs** | Base | - | 3.00 ± 0.00 | **3.62 ± 0.16** | 3.67 ± 0.14 | 3.00 ± 0.14 | 4.53 ± 0.12 | 1.67 ± 0.29 |
| | ↳ CoT | - | 3.53 ± 0.12 | 3.09 ± 0.08 | 3.33 ± 0.14 | 3.43 ± 0.00 | 3.80 ± 0.20 | 1.83 ± 0.29 |

Table 1: Recall (see §5.3) of known failure patterns (meta-labels) with generations for MathCAMPS and MMLU, under an error-ratio cutoff of 0.5. The columns show the LLM whose failure patterns are assessed for a particular dataset, while rows show the different methods. CoT indicates that the method is provided the LLM's chain of thought in addition to the wrong instances as described in §5. The numbers in parentheses show the count of known failure patterns for each LLM–dataset pair. Results are averaged across three runs of the o3-mini scorer.

Two erroneous mapper nodes are merged if they are connected by an edge and the resulting node yields a higher error score. We then select the top $k$ clusters (merged nodes) based on the error score and use the prompt from the `Direct` method to generate a single failure pattern for each cluster; this process adapts the recent work (Yan et al., 2025a) that utilizes an explainer agent for a set of mapper nodes; see Appendix D for details.

## 5.3 Evaluation Measurement

For each LLM, we generate candidate failure patterns using the methods above and compare them to the known failure patterns based on meta-label groups (standards for MathCAMPS, subjects for MMLU; thresholded with an error-ratio above 0.5). Recall is computed by scoring each generated pattern against all ground-truth patterns, taking the highest match, and averaging across all ground-truth patterns. To score the match between a generated description and a ground-truth pattern, we develop an LLM-as-a-judge (Zheng et al., 2023). Specifically, we prompt `o3-mini-2025-01-31` to rate how closely the generated and ground-truth descriptions align; see App. B for the exact prompt. The rating is on a scale from 1 to 5: 1 (No Match), 2 (Weak Match), 3 (Moderate Match), 4 (Strong Match), and 5 (Perfect Match). See Appendix C for the examples of the o3-mini's ratings.

To assess the validity of using the o3-mini model as a judge, we analyze the agreement between its scores and those of a human judge for the `Direct` method. The mean absolute difference between the scores given by the human judge and o3-mini is only 0.051. We also calculate the weighted Cohen's kappa (Cohen, 1968) between the human and o3-mini scores. This metric accounts for the relative differences between scores, penalizing larger disagreements. Cohen's kappa scores of 0.838 for quadratic weights and 0.744 for linear weights indicate high agreement.

## 5.4 Results

Table 1 shows that the relatively simple `Direct` prompting method reaches recall scores close to 4 (Strong Match). In other words, it nearly recovers the patterns accurately.[4] By contrast, `IntegrAI` reaches only 3.25 at best, suggesting that noise in its generated failure patterns may have contributed to the ineffectiveness of teaching observed by Mozannar et al. (2023). Since the overall effectiveness of the AI-integration teaching

---

[4]Because a perfect score of 5 is rare, we manually checked generation scores (see App. C). We find that, although not perfect, a score of 4 generally describes the pattern in sufficient detail. In contrast, patterns scoring 3 often omit key aspects, making their instructional value uncertain and sometimes misleading, while those scoring 2 diverge too far from the ground truth to be useful for teaching.

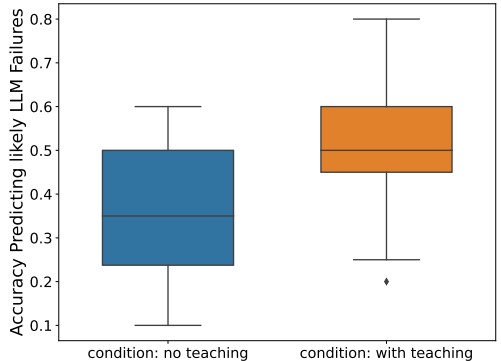

| Question Subject | No Teaching | Teaching | Δ |
|---|---|---|---|
| global_facts | 0.13 | 0.87 | 0.74 |
| no-match_failure_pattern | 0.00 | 0.63 | 0.63 |
| sociology | 0.33 | 0.67 | 0.34 |
| elementary_mathematics | 0.60 | 0.92 | 0.32 |
| logical_fallacies | 0.36 | 0.64 | 0.28 |
| high_school_psychology | 0.47 | 0.73 | 0.27 |
| public_relations | 0.24 | 0.47 | 0.23 |
| marketing | 0.50 | 0.67 | 0.17 |
| business_ethics | 0.67 | 0.83 | 0.16 |
| high_school_government_and_politics | 0.50 | 0.63 | 0.13 |
| high_school_computer_science | 0.33 | 0.33 | 0.00 |
| anatomy | 0.50 | 0.40 | -0.10 |
| high_school_statistics | 0.50 | 0.31 | -0.19 |
| machine_learning | 0.50 | 0.08 | -0.42 |
| moral_scenarios | 0.64 | 0.12 | -0.52 |

Figure 4: **Left:** User accuracy in predicting LLM failures before vs. after learning failure patterns. **Right:** The effect of teaching on users' accuracy in predicting LLM failures for different subjects.

pipeline depends on the quality of this stage, these results indicate that Stage 2 remains a key bottleneck and that improving it could strengthen the entire pipeline.

In addition, we highlight observations with practical importance for researchers developing the AI-integration teaching pipelines. First, we find no connection between recall and the number of patterns to be recognized. That is, having more known failure patterns to discover and describe does not necessarily make the second stage of the pipeline more difficult. Second, a notable variation in recall of known failure patterns across LLMs — regardless of whether the number of patterns is the same or not — supports our expectation that the choice of the underlying LLM is critical for fair comparison and rigorous evaluation when studying these pipelines. For example, reporting only the performance of the `Direct` method for Claude-3-Opus's errors would overstate its effectiveness compared to the full results in Table 1. Future work could explore how the composition of Stage 1 data affects performance in Stage 2. It appears that the number of failure patterns alone is not the determining factor, contrary to our initial speculation.

Finally, providing CoTs in addition to error instances for MathCAMPS shows mixed results in terms of improving a method's recall of known failure patterns. Upon manual review, we observe that in settings where adding CoTs decreased recall (e.g., `D5`, Opus), the descriptions appeared to more strongly feature general statements about the level of complexity or number of steps required to solve the problems instead of concepts.

# 6 Measuring Teaching Effectiveness

The AI-integration teaching pipeline aims to help users better integrate AI predictions into decision-making, but how can we tell if teaching was effective? This section examines how different metrics can affect conclusions about teaching success.

In prior work, Mozannar et al. (2023) used human-AI team accuracy to assess the effectiveness of Stage 3 of the AI-integration teaching pipeline, finding no improvement in accuracy for MMLU. From this, they conclude that the pipeline is unsuccessful in the LLM setting. However, it is unclear that successful teaching must lead to an increase in team accuracy. For example, a user might defer to a model for low-confidence tasks, even when aware of its likeliness to make a mistake.

Instead of team accuracy, we propose to measure whether users can predict when an LLM is likely wrong. This metric does not require users to answer the question themselves, but only to determine if they should use an LLM to get the answer.

**Setup.** To check whether this metric yields a different interpretation of the success of Stage 3, we repeat Mozannar et al. (2023)'s two-part user study for MMLU with minor modifications to allow for our new metric. Initially, users are presented 20 multiple-choice questions and asked to record whether they would use AI, would not use AI, or are uncertain, as well as the reasoning behind their choice. Next, users undergo a

training stage where they are taught descriptions of the model's strengths and weaknesses. After teaching, users are tested again on the same 20 questions. Since our aim is to compare to prior work, we use the same questions and automatically generated failure patterns from Mozannar et al. (2023). Additionally, we add 5 questions to each set which do not match any of the taught failure patterns. These questions allow us to measure whether users apply the learned guidelines beyond when they should.

Screenshots of the user interface are available in Appendix F. Initially, we sought to recruit crowdworkers for this study through Prolific, however, examining the free response explanations given during a pilot study with 6 users revealed a mix of poor-faith attempts (nonsensical reasoning) and failure to follow task instructions (e.g., assuming access to other resources besides the LLM prediction). We attempted to resolve the latter with rewrites of the instructions, but this only resulted in 2/6 responses where we trusted the participants' understanding of the task. In the end, we decided to recruit users through our personal network (NLP doctoral students and their friends) in order to more easily communicate the intent of the study and resolve questions regarding task instructions. In total, we recruited 20 participants with the only exclusion criterion being fluency in English. We obtained IRB approval from our institution and informed consent from all participants. Each participant answered a randomized ordering of the same 20 questions in two separate rounds for a total of 800 annotations. Each was paid $15 and completed the study within 1 hour and 15 minutes, resulting in a minimum pay rate of $12/hour and a total study cost of $390.

**Results.** For 18/20 participants, we observe an increase in our new metric after teaching. Fig. 4 (Left) shows the aggregate performance difference, and Fig. 6 (Appendix) the differences for individual participants. The Shapiro-Wilk test (Shapiro & Wilk, 1965) indicates that user performances in both studies follow a normal distribution, and hence, we run a paired t-test. This demonstrates that teaching has a significant positive effect on user performance, with a p-value of 0.0009. To assess the effect size, we calculate Cohen's d (Cohen, 1988), which measures the standardized difference between the performance means of the two studies. The resulting value of 0.88 suggests a large effect size.

Together, these results demonstrate that teaching has a positive impact for MMLU, contrary to Mozannar et al. (2023)'s study using human-AI team accuracy. The benefits of teaching were more pronounced among participants without graduate-level computer science backgrounds (0.20 vs. 0.11), suggesting that users less familiar with LLMs stand to benefit more from teaching. However, even after teaching, participants' understanding of when to use LLMs could still be as low as 25% (see App. Fig. 6), so there remains notable room for improving teaching effectiveness.

To qualitatively understand what users learned from teaching, we analyzed its effect with respect to subjects in MMLU. Fig. 4 (Right) shows that teaching improved user accuracy in anticipating LLM failures for questions from 10 subjects, while for 4 subjects accuracy declined. This could be explained by varying difficulty in applying descriptions of failure patterns to questions. With the exception of `business_ethics`, all of the subjects where we observe improvement from teaching are specifically mentioned in the taught descriptions. In contrast, for subjects where performance declined, questions appear to have been mistakenly matched to descriptions which don't clearly apply. This suggests that matching instances to failure descriptions is non-trivial and can impact teaching effectiveness. More details of this analysis are provided in App. E.

## 7 Related Work

**Automated Error Discovery.** Identifying systematic failures and error types of AI models has an extensive history, including manual analysis (Vasudevan et al., 2022; Zheng et al., 2024), automatic methods for slice discovery (Eyuboglu et al., 2022; d'Eon et al., 2022; Hua et al., 2023), and building interactive systems for debugging models (Wu et al., 2019; Rajani et al., 2022; Yan et al., 2025b). Some of these methods leverage embeddings to form clusters representing systematic model failures and then describe them (Eyuboglu et al., 2022; d'Eon et al., 2022; Mozannar et al., 2023; R Menon & Srivastava, 2024), while others use prompting (Zhong et al., 2022; 2023). To generate failure descriptions, LLMs are used to describe the features common among examples in a cluster. Descriptions are sometimes further refined by comparing and contrasting examples from inside and outside of the cluster (Mozannar et al., 2023; R Menon & Srivastava, 2024). In

general, methods like these are candidates for producing and describing error groups in Stage 2 of the AI-integration teaching pipeline. We explore a diverse set of these methods in §5.

**Onboarding.** A number of studies have proposed to apply a training or onboarding procedure to help people work better with AI systems (Mozannar et al., 2023; Cabrera et al., 2023; Mozannar et al., 2022; Lai et al., 2022; 2020). Closest to our work is Mozannar et al. (2023) which is the only work which incorporates all of the stages in the AI-integration teaching pipeline. The failure of this setup for LLMs motivates our formalization and analysis of the AI-integration teaching pipeline.

**Error Predictability.** Recent research has highlighted how humans struggle to predict when an AI system is likely to make an error based on the task instance, prompt, or other information about individual model input/output (Green & Chen, 2019; Lai & Tan, 2019; Lai et al., 2020; Zhang et al., 2020). In this work, we propose to use error predictability instead of human-AI team accuracy, to better measure the effectiveness of teaching users the failure patterns of LLMs.

**Overreliance.** The overreliance of users on AI predictions is a well-known and persistent challenge (Passi & Vorvoreanu, 2022). Although users are often provided with explanations of model outputs, these can lead users to trust model predictions when they should not (Bansal et al., 2021). Prior work has confirmed this effect and proposed mitigations, including contrastive explanations (Si et al., 2024), designing explanations to reduce cognitive cost (Vasconcelos et al., 2023), and augmenting explanations with source attributions (Kim et al., 2025). In contrast to this line of work, we examine the feasibility of mitigating overreliance by teaching users to recognize a model's failure patterns before using it for their task.

## 8  Conclusions & Future Directions

We study where a proposed approach for mitigating overreliance on LLMs, the AI-integration teaching pipeline, falls short. First, we define metrics we term coverage and error ratio, and with them show that across several LLM–dataset pairs there do exist patterns describing sizable groups where the LLM is error-prone (RQ1). Their absence is therefore not what limits the pipeline's effectiveness. Second, we define a recall-based metric for Stage 2, and find that this stage is the weak link: current methods for discovering and describing these patterns need improvement (RQ2). Third, we show that the pipeline appears effective when evaluated by whether users can recognize when not to use an LLM, unlike when judged by AI-assisted human accuracy (RQ3).

These results suggest that the pipeline could be a viable approach to addressing overreliance on LLMs and merits further research. Promising directions include explaining why prompting-based methods outperform embedding-based ones and developing stronger Stage 2 methods, both for discovering and describing failure patterns and for mapping instances to the generated properties. Future work should also test Stage 2 with more complex patterns, develop evaluation beyond retrieval of known patterns, and examine how different error-ratio and coverage thresholds affect teaching outcomes. Finally, an open challenge is how to guide users who recognize they should not use an LLM, yet cannot complete the task on their own.

## 9  Limitations

While our work provides a systematic analysis of the AI-integration teaching pipeline, we acknowledge certain limitations of the work with respect to individual stages of the pipeline.

In Stage 1, we use meta-labels to identify failure patterns in two representative datasets: MMLU, chosen because it is central to the prior work we focus on, and MathCAMPS, selected because its meta-labels (educational standards) differ notably from MMLU's (subjects). Although these datasets are a reasonable testbed for our analysis, additional datasets and tasks would further confirm that failure patterns *frequently* exist in practice and that Stage 1 is *generally* not the limiting factor.

In Stage 2, we show that a prompting method can generate somewhat accurate descriptions of known failure patterns. Further analysis could help explain why it is more successful, but devising a research plan for such analysis is far from straightforward. Moreover, although we provide the first evaluation of Stage 2, it assesses

methods only by how well their generations match known failure patterns. We do not yet have an automatic way to compare methods based on the full set of patterns they produce. This is an open challenge for future work rather than a simple extension.

Further, we do not analyze different implementations of Stage 3 in the pipeline. The interface and method used to most effectively teach people about model failure patterns is a rich area for future work.

In Stage 4, we carry out a user study to illustrate the importance of the choice of metric in measuring teaching effectiveness. We run the study for MMLU in order to compare directly to prior work. Additional user studies on MathCAMPS and other datasets would further strengthen our findings. However, even for MMLU we faced practical challenges in recruiting reliable annotators. Additionally, our 20-participant study costs $390. Our findings would benefit from a larger sample size, which could also enable additional analyses targeting how aspects of a user's background beyond familiarity with LLMs (which we analyzed) might influence teaching effectiveness.

Finally, the scope of this work is intentionally limited to analyzing the existing AI-integration teaching pipeline to understand its weaknesses. Proposing alternative pipelines goes beyond our research questions. Future alternatives could be designed following our findings.

## 10 Ethical Considerations

In this work, we study a pipeline for training humans to integrate LLM predictions to solve tasks. While the goal is to affect human decision-making in positive ways like reducing overreliance, there are inherent risks. It is possible that the failure patterns taught to users could lead them to integrate LLM predictions in unexpected ways that could cause them to rely on LLMs inappropriately. This highlights the importance of robust understanding of the different stages of the AI-integration teaching pipeline and the factors that could cause it to fail. For example, if the dataset used in Stage 1 does not contain examples of a particular failure pattern of an LLM, then it cannot possibly be discovered and taught to users in later stages. Similarly, if a method for generating failure patterns in Stage 2 struggles to accurately describe certain concepts, this could also lead to biased teaching. In Stage 3, different approaches to teaching could advantage users with certain backgrounds over others. By formalizing and analyzing the AI-integration teaching pipeline, we hope that this work contributes towards a better understanding of these various failure modes and ultimately helps users to rely appropriately on LLMs.

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

# A   Prompts for Generating Error Types

Listing 1: Prompt for generating error types with the Direct method when number of error types is specified.

```
I am a user of an AI system. My goal is to figure out what types of questions the AI
    model is more likely to fail on and teach these principles to other users of the
    system.
In other words, I want to discover what specific features of the text distinguish the
    questions where AI is correct from the questions where AI is wrong.
Given a list of instances where an AI was incorrect, please generate the {num_hyps}
    most salient hypotheses which describe instances where AI is more likely to be
    wrong.
Format the response as a json object.
For example, here is how the output would be formatted for 3 generated hypotheses.
{{"hypothesis0": "requires addition with two-digit numbers",
"hypothesis1": "contains negation words in the question",
"hypothesis2": "involves computing an integral as an intermediary step"}}

There are {num_hyps} salient error types, so you should generate {num_hyps}
    hypotheses.
The description(s) should be concise, understandable to humans, and describe as many
    of the wrong instances as possible. Ideally, the list would cover all of the
    datapoints. Avoid using subjective or vague language such as "complex" or
    "ambiguous" and focus on specific characteristics. If there are no distinctive
    properties then list "None" as the entry.

Error Instances:
***
{error_instances}
***

Based on the error instances from the above, the model is likely to make an error
    when a question
```

Listing 2: Prompt for generating error types with the Direct method when number of error types is **unspecified.**

```
I am a user of an AI system. My goal is to figure out what types of questions the AI
    model is more likely to fail on and teach these principles to other users of the
    system.
In other words, I want to discover what specific features of the text distinguish the
    questions where AI is correct from the questions where AI is wrong.
Given a list of instances where an AI was incorrect, please generate the most salient
    hypotheses which describe instances where AI is more likely to be wrong.
Format the response as a json object.
For example, here is how the output would be formatted for 3 generated hypotheses.
{{"hypothesis0": "requires addition with two-digit numbers",
"hypothesis1": "contains negation words in the question",
"hypothesis2": "involves computing an integral as an intermediary step"}}

You should generate as many hypotheses as you deem fit to cover the most salient
    error types.
The description(s) should be concise, understandable to humans, and describe as many
    of the wrong instances as possible. Ideally, the list would cover all of the
    datapoints. Avoid using subjective or vague language such as "complex" or
    "ambiguous" and focus on specific characteristics. If there are no distinctive
    properties then list "None" as the entry.

Error Instances:
```

```
***
{error_instances}
***

Based on the error instances from the above, the model is likely to make an error
    when a question
```

Listing 3: Prompt for generating error types with D5. The number of error types is specified as a hyperparameter which is used to iteratively refine a large pool of candidate descriptions. The same prompt is used for MathCAMPS and MMLU except for the description of the domain denoted in curly braces

```
"prompt":
Group A:
{error_instances}

Group B:
{correct_instances}

The dataset includes {elementary level math problems for grades K-8, questions
    related to math topics testing different educational standards, questions
    related to health topics}. The two groups are generated based on whether the
    AI answered correctly or not. The Group A snippets are questions where the AI
    system was wrong, while the Group B snippets are questions where the AI system
    was correct.

I am user of this AI system. My goal is to figure out what types of questions the
    AI model is more likely to fail on and teach these principles to other users
    of the system.
In other words, I want to discover what specific features of the text distinguish
    the questions where AI is correct from the questions where AI is wrong.
    \n\nPlease write a list of the most salient hypotheses about the datapoints
    from Group A (listed by bullet points \"-\"). Each hypothesis should be
    formatted as a sentence fragment. Here are three examples.\n- \"requires
    addition with two-digit numbers\"\n- \"contains negation words in the
    question\"\n- \"involves computing an integral as an intermediary step\"

Based on the two sentence groups (A and B) from the above, more sentences in
    Group A",
```

## B Prompts for Ranking a Match Between a Generated Description and a Proxy Error Type

Listing 4: Prompt for rating a match between a generated description and the reference MathCAMPS standard. This prompt was developed with assistance from Claude.

```
# Instructions for Evaluating Math Problem Description Similarity

When evaluating how closely a submitted description matches the reference
    description, consider both the core mathematical concepts and the specific details
    mentioned.

## Rating Scale
1. **Perfect Match (5/5)**: Description captures all key mathematical concepts and
    specific details from the reference.
2. **Strong Match (4/5)**: Description captures the main mathematical concepts but
    may miss minor details.
```

```
3. **Moderate Match (3/5)**: Description captures some key concepts but misses
     significant details or uses imprecise terminology.
4. **Weak Match (2/5)**: Description only vaguely relates to the reference, missing
     most key concepts.
5. **No Match (1/5)**: Description fails to capture any relevant mathematical
     concepts from the reference.

## Evaluation Process
1. Identify the core mathematical concepts in the reference description (both long
     and short versions)
2. Determine which specific details or applications are essential
3. Compare the submitted description against these elements
4. Assign a rating based on how completely the core concepts and essential details
     are captured

## Example Evaluation
**Reference (Long)**: "Solve real-world and mathematical problems involving the four
     operations with rational numbers."
**Reference (Short)**: "Add/sub/mult/div with fractions"

**Core Concepts**:
- Arithmetic operations (addition, subtraction, multiplication, division)
- Fractions/rational numbers

**Rating Examples**:
- **Perfect (5/5)**: "Problems with the four arithmetic operations with fractions to
     solve problems."
- **Strong (4/5)**: "Problems with fractions using different operations to solve math
     problems."
- **Moderate (3/5)**: "Solving problems with fractions" (missing specific mention of
     operations)
- **Weak (2/5)**: "Working with number operations" (mentions operations but not
     fractions)
- **No Match (1/5)**: "Solving geometry problems" (unrelated mathematical domain)

When judging, remember to look for both the mathematical content (operations, number
     types, etc.) and number types and ranges (whole numbers, fractions, decimals,
     etc.).

Structure your entire output in JSON format as follows:\n" \
               "{\n" \
               "    'Additional Comments': [comments],\n" \
               "    'Final Rating': [verdict]\n" \
               "}\n" \
               "where \"verdict\" must be 1, 2, 3, 4, or 5 nothing else.
```

Listing 5: Prompt for rating a match between a generated description and the reference MathCAMPS standard.

```
# Instructions for Evaluating Math Problem Description Similarity to MMLU-Math Topics

When evaluating how closely a submitted description matches the reference topic from
     MMLU-math, consider how well the description aligns with the mathematical domain
     and concepts typically covered under that topic.

## Rating Scale

1. Perfect Match (5/5): Description directly addresses the core mathematical concepts
     and methods central to the topic.
```

```
2. Strong Match (4/5): Description captures the main mathematical domain and most
     relevant concepts for the topic.
3. Moderate Match (3/5): Description relates to the topic but may be too broad,
     narrow, or miss some key conceptual areas.
4. Weak Match (2/5): Description only tangentially relates to the topic, touching on
     peripheral concepts.
5. No Match (1/5): Description addresses a completely different mathematical domain
     or unrelated concepts.

Evaluation Process

1. Identify the core mathematical domain and key concepts typically associated with
     the reference topic
2. Consider the scope and typical applications within that mathematical area
3. Compare the submitted description against the expected conceptual coverage
4. Assign a rating based on how well the description aligns with the topic's
     mathematical focus

## Example Evaluation
Reference Topic Description: "Algebra"
Expected Core Concepts:

Algebraic expressions and equations
Variable manipulation and solving
Functions and their properties
Polynomial operations

Rating Examples:

Perfect (5/5): "Solving algebraic equations and working with variables to find
     unknown values"
Strong (4/5): "Problems involving equations and mathematical expressions with
     unknowns"
Moderate (3/5): "Mathematical problem-solving with symbols" (too broad, lacks
     specificity)
Weak (2/5): "Working with mathematical formulas" (vague, could apply to many areas)
No Match (1/5): "Calculating areas and perimeters of geometric shapes" (geometry, not
     algebra)

## Additional Considerations

Consider whether the description captures the appropriate level of mathematical
     sophistication for the topic
Look for key terminology and concepts that are characteristic of the mathematical
     domain
Evaluate if the scope is appropriately matched (not too broad or too narrow)

Structure your entire output in JSON format with the following entries:
'Additional Comments': [comments],
'Final Rating': [verdict]
where "verdict" must be 1, 2, 3, 4, or 5 nothing else.
```

Listing 6: Prompt for rating a match between a generated description and the reference MMLU Health topic.

```
# Instructions for Evaluating Health Problem Description Similarity to MMLU-Health
     Topics
When evaluating how closely a submitted description matches the reference topic from
     MMLU-health, consider how well the description aligns with the medical/health
     domain and concepts typically covered under that topic.
```

## Rating Scale

1. Perfect Match (5/5): Description directly addresses the core health concepts, medical principles, or clinical scenarios central to the topic.
2. Strong Match (4/5): Description captures the main health domain and most relevant concepts for the topic.
3. Moderate Match (3/5): Description relates to the topic but may be too broad, narrow, or miss some key clinical/health areas.
4. Weak Match (2/5): Description only tangentially relates to the topic, touching on peripheral health concepts.
5. No Match (1/5): Description addresses a completely different health domain or unrelated concepts.

## Evaluation Process

1. Identify the core health/medical domain and key concepts typically associated with the reference topic
2. Consider the scope and typical clinical applications within that health area
3. Compare the submitted description against the expected conceptual coverage
4. Assign a rating based on how well the description aligns with the topic's health/medical focus

Example Evaluation
Reference Topic: "Cardiology"
Expected Core Concepts:

Heart anatomy and physiology
Cardiovascular diseases and conditions
Diagnostic procedures (ECG, echocardiograms)
Treatment approaches and medications
Risk factors and prevention

## Rating Examples:

Perfect (5/5): "Diagnosing and treating heart conditions including arrhythmias and coronary artery disease"
Strong (4/5): "Medical problems related to heart function and cardiovascular health"
Moderate (3/5): "Health issues affecting the circulatory system" (too broad, lacks clinical specificity)
Weak (2/5): "Managing chronic health conditions" (vague, could apply to many specialties)
No Match (1/5): "Treating skin disorders and dermatological conditions" (dermatology, not cardiology)

## Additional Considerations

Consider whether the description captures the appropriate level of medical sophistication for the topic
Look for key terminology and concepts that are characteristic of the health domain
Evaluate if the scope appropriately matches clinical practice areas (not too broad or too narrow)
Consider both theoretical knowledge and practical clinical applications

Structure your entire output in JSON format with the following entries:
'Additional Comments': [comments],
'Final Rating': [verdict]
where "verdict" must be 1, 2, 3, 4, or 5 nothing else.

## C   o3-mini Evaluation Traces

Below we show example evaluation traces from the o3-mini scorers. Table 2 shows examples of generated descriptions achieving scores of 4,3,2 for meta-labels from MathCAMPS, **MMLU** Math and **MMLU** Health.

Listing 7: Trace of a o3-mini rating a match between a generated description and the reference standards for Claude-3-Opus (4). The Group description is evaluated with respect to the reference description for each standard and the max score is reported.

```
Group description: - "requires addition of fractions"

Reference detailed description: Solve real-world and mathematical problems involving
    the four operations with rational numbers.

Reference gist: Add/sub/mult/div with fractions

Comments: ['The submitted group description only mentions addition of fractions,
    while the reference entails solving problems using all four operations (addition,
    subtraction, multiplication, and division) with rational numbers.']

Rating: 3
Group description: - "requires addition of fractions"

Reference detailed description: Use the four operations to solve word problems
    involving distances, Intervals of time, liquid volumes, masses of objects, and
    money, including problems involving simple fractions or decimals, and problems
    that require expressing measurements given in a larger unit in terms of a smaller
    unit. Represent measurement quantities using diagrams such as number line diagrams
    that feature a measurement scale.

Reference gist: Word problems with fractions

Comments: The submitted description only mentions addition of fractions, while the
    reference involves solving real-world word problems using all four operations
    (including addition) with fractions and decimals, as well as applying these
    operations in contexts like distances, time intervals, liquid volumes, masses, and
    money, and sometimes using diagrams. The provided description captures only a
    small aspect of the reference.

Rating: 2
Group description: - "requires addition of fractions"

Reference detailed description: Add and subtract fractions with unlike denominators
    (including mixed numbers) by replacing given fractions with equivalent fractions
    in such a way as to produce an equivalent sum or difference of fractions with like
    denominators.

Reference gist: Add/sub fractions

Comments: The group description only mentions the addition of fractions, whereas the
    reference detailed description includes both addition and subtraction (with unlike
    denominators, including mixed numbers, and the method of converting fractions to
    equivalent fractions to achieve like denominators). Thus, while the group
    description captures a core aspect (addition of fractions), it misses significant
    parts of the reference.

Rating: 3
Group description: - "requires addition of fractions"
```

```
Reference detailed description: Apply and extend previous understandings of addition
    and subtraction to add and subtract rational numbers; represent addition and
    subtraction on a horizontal or vertical number line diagram.

Reference gist: Add/sub with fractions

Comments: The group description only mentions addition of fractions, omitting
    subtraction and the representation on a number line that are present in the
    detailed reference description. While it captures part of the core concept (adding
    fractions), it leaves out significant elements.

Rating: 3
Max rating for this hypothesis: 3
```

| Meta-Label | Generated Failure Pattern | Method | Score |
|---|---|---|---|
| Abstract Algebra | requires understanding or manipulating advanced mathematical concepts or structures, such as field extensions, eigenvectors, or cyclic groups | Direct | 4 |
| | The region focuses on complex mathematical problem-solving and logical reasoning, distinct from basic arithmetic or non-mathematical scenarios. | IntegrAI | 3 |
| | asks for exact solutions to algebraic problems involving multiple unknowns or simultaneous equations | Direct | 2 |
| Virology | – | – | 4 |
| | requires understanding of biological or biochemical mechanisms | D5 | 3 |
| | requires knowledge of specific medical or biological terms | Direct | 2 |
| 7.NS.A.2 ("Mult/div with fractions") | contains a mix of fractions and whole numbers in multiplication or division | Direct | 4 |
| | requires careful interpretation of word problems involving sequential operations on fractions | Direct | 3 |
| | requires operations involving complex fraction arithmetic, especially addition and subtraction | Direct | 2 |

Table 2: Examples of generated failure patterns and their matching score with meta-labels from MMLU Math, MMLU Health, and MathCAMPS. For Virology, no method achieved a score of 4

## D  Method Hyperparameters and Details

**D5.**  Originally, D5 was introduced for the more general task of discovering the differences between two arbitrary sets of text corpora. We take advantage of the flexibility of this method and adapt it the task of generating failure patterns by setting the text groupings that are contrasted to be the set of instances a model got wrong and correct, respectively. We use Zhong et al. (2023)'s open-sourced implementation of D5 and generate descriptions for each setting using the same basic prompt structure, filling in the appropriate template variables to match the dataset we are using (e.g. specifying the domain to be K-8 math questions or health questions). The only modification we make is to change the model which generates candidate descriptions to be `gpt-4o-2024-08-06` for fair comparison with other methods. We also update the validation function to use `gpt-4o-2024-08-06` instead of the fine-tuned T5 validator. This validation step classifies at

the instance level as to whether or not a candidate description applies to a hypothesis for a sample from both groups of text. These scores are then aggregated to determine an overall score for the candidate. Since the final number of candidates to generate is a hyperparameter, we set it equal to the number of proxies above the error threshold and only evaluate for the case where the number of error types is specified. We allow a max number of 20 hypotheses to be generated and enable the early stopping parameter. For all other hyperparameters we use the default settings.

**IntegrAI.** To discover error groups with IntegrAI we follow guidance in Mozannar et al. (2023)'s repo and use the single-model setting, the `all_0` prior, and a loss threshold of 0.5. This sets the assumption that AI performs better than the loss threshold on all instances, allowing the algorithm to discover clusters that contradict it i.e. error clusters. We set the parameters for cluster size as min size $= 0.03$ and max size $= 0.2$ to ensure that the discovered groups achieve reasonable coverage. To ensure that discovered regions achieve the minimum error-ratio threshold of 0.5 we set consistency=0.33. Since this method requires the number of regions to discover to also be specified, we set this equal to the number of ground truth proxies above the error-ratio threshold and evaluate models only for the case where the number of error groups is specified. In order to ensure a fair comparison with other methods, we also update the description generation portion of IntegrAI to use `gpt-4o-2024-08-06`.

**Mapper Graphs.** For each dataset, we first construct a mapper graph following the parameter setup described in (Yan et al., 2025a). Specifically, we use the $L_2$-norm as the filter function, divide its range into 100 intervals with 50% overlap, and apply DBSCAN for clustering (with $\epsilon$ determined by the elbow method and min_samples set to 3).

An erroneous mapper node is one that contains wrong instances, characterized by its error score. At each iteration, we merge a pair of mapper nodes (clusters) connected by an edge as long as the resulting node (cluster) has a higher error score. Following a greedy strategy, we keep merging until the error score no longer increases. This process results in $m$ error regions, each representing a connected mapper subgraph. To extract a specified number of error regions $k$, we simply return the top $k$ clusters (merged nodes) ranked by their error scores. If $k > m$, we return all $m$ available regions. Based on the explainer agent provided by the Explainable Mapper (Yan et al., 2025a), we input all wrong instances from each error region and prompt `GPT-4o-2024-08-06` to generate a single description of the error. The corresponding prompt is provided in Appendix A.

## E   Additional Human Study Results

Fig. 5 shows the distribution of score differences for users after they are taught about LLM failure patterns while Fig. 6 visualizes the performance of individual participants.

**Effects of Teaching by Subject**   In Fig. 4, we observe that teaching helped notably for questions belonging to subjects such as `global_facts`, `sociology`, and `elementary_mathematics`, suggesting that the descriptions of failure patterns clearly conveyed these topics to users. In fact, all of the topics where we observe improvement are specifically mentioned in the descriptions, except for `business_ethics`. However, we also notice that for 4 subjects the average accuracy declined. While none of these subjects are mentioned directly in the descriptions, some could reasonably be interpreted as matching them. For example, users might associate `moral_scenarios` with the description *"questions related to sociology. . . "*. Since users learned that LLMs tend to succeed on questions matching this description, they could mistakenly apply this same takeaway to questions about `moral_scenarios`. Our results support this hypothesis. Before teaching, users predicted the LLM to fail on 16 questions from this topic; afterward, that number fell to just 3. Similarly, for `machine_learning` and `high_school_statistics`, the number of questions where the LLM is predicted as likely to succeed drops from 7 to 2 and 16 to 10, respectively. This could be due to users mapping questions from these topics to the description *"questions related to math and involving quantitative calculations. . . "* which has the takeaway that LLMs are likely to fail. Interestingly, humans showed no difference after teaching for questions from `high_school_computer_science`, suggesting that teaching did not alter their perception of an LLM's ability for questions from this topic. This is reasonable since none of the taught descriptions

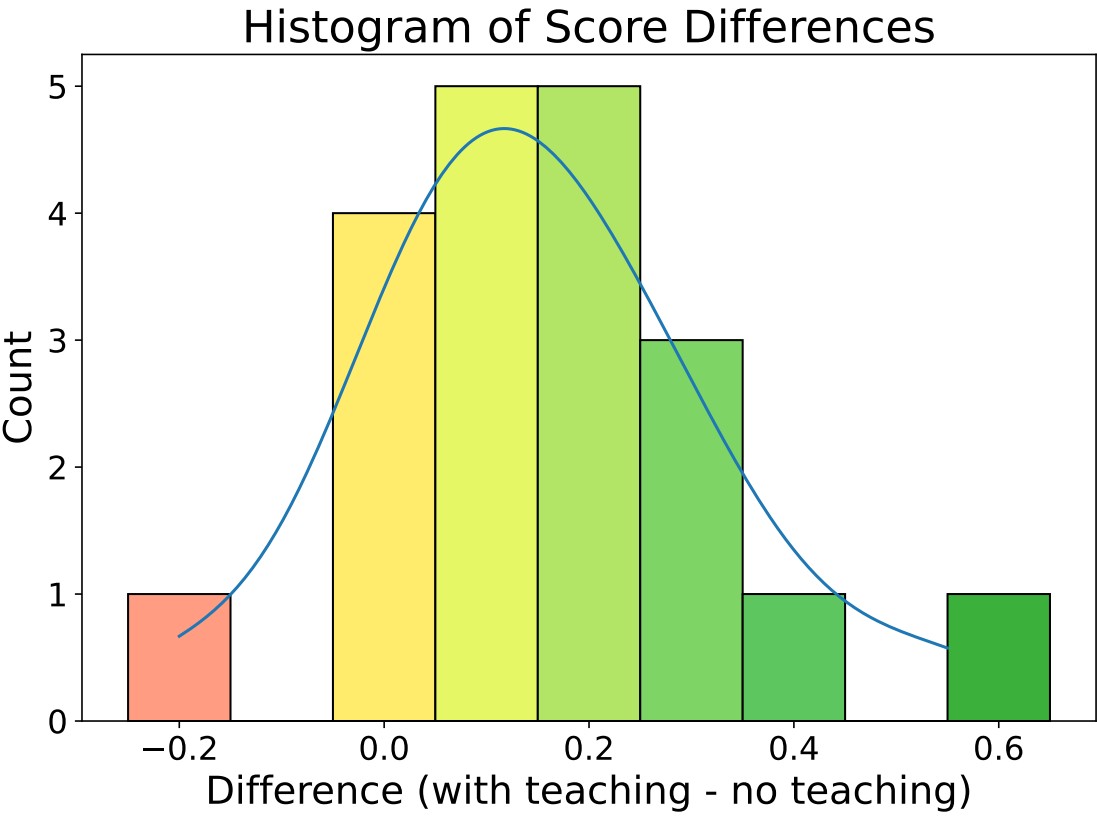

Figure 5: Distribution of differences between users' accuracy in predicting LLM failures after being taught failure patterns.

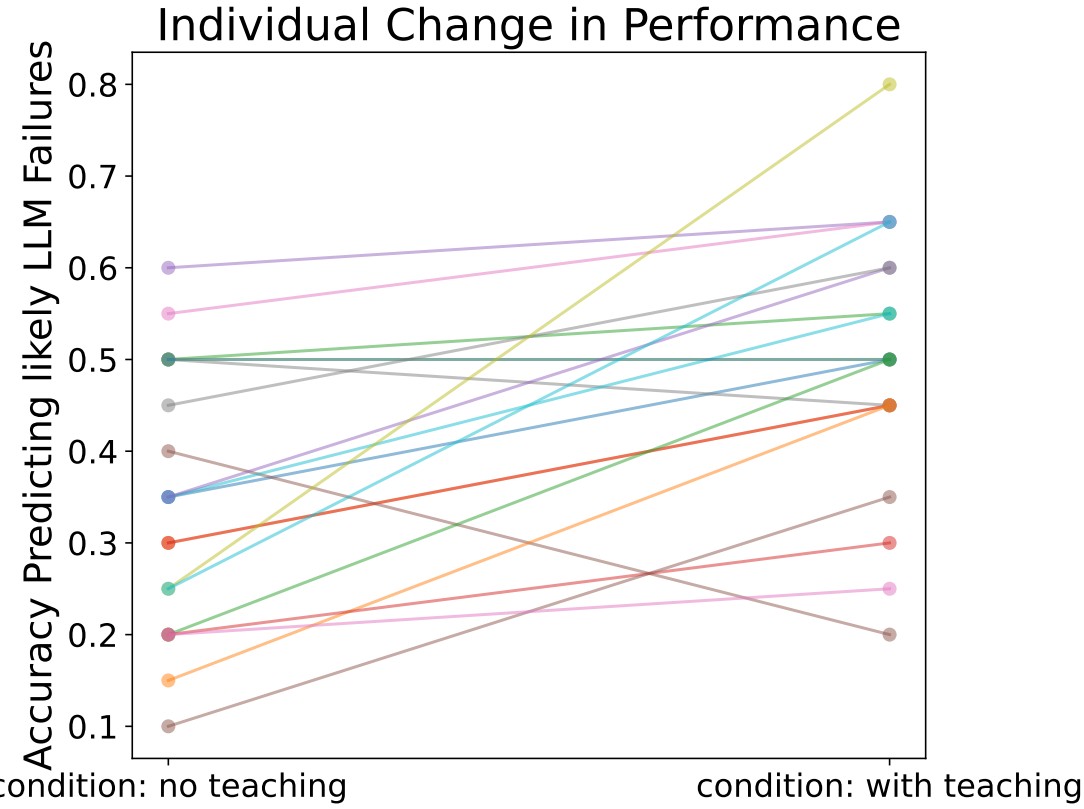

Figure 6: Individual user performance changes before and after teaching.

mention computer science and there are no obvious alternative descriptions. Finally, for questions which do not match any of the taught failure patterns, users can only recognize that there is no applicable description with an average accuracy of 63%.

## F    Human Study Interface

In Figures 7–9, we show screenshots from the user study interface. Templates are released at (anonymizedforreview)

## G    Error-Ratio and Coverage for Additional Meta-Label Groups

Figure 10 shows coverage and error-ratio for other subcategories in MMLU. Meta-labels for subcategories such as philosophy, law, physics, chemistry align with our criteria for instance groups worth generating failure patterns for, while others such as history and biology do not.

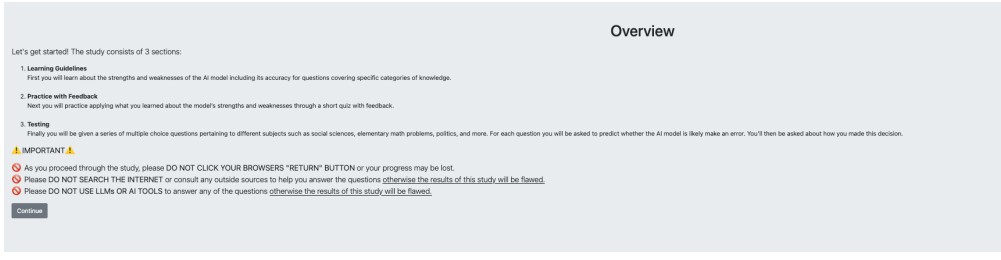

(a) Initial overview instructions.

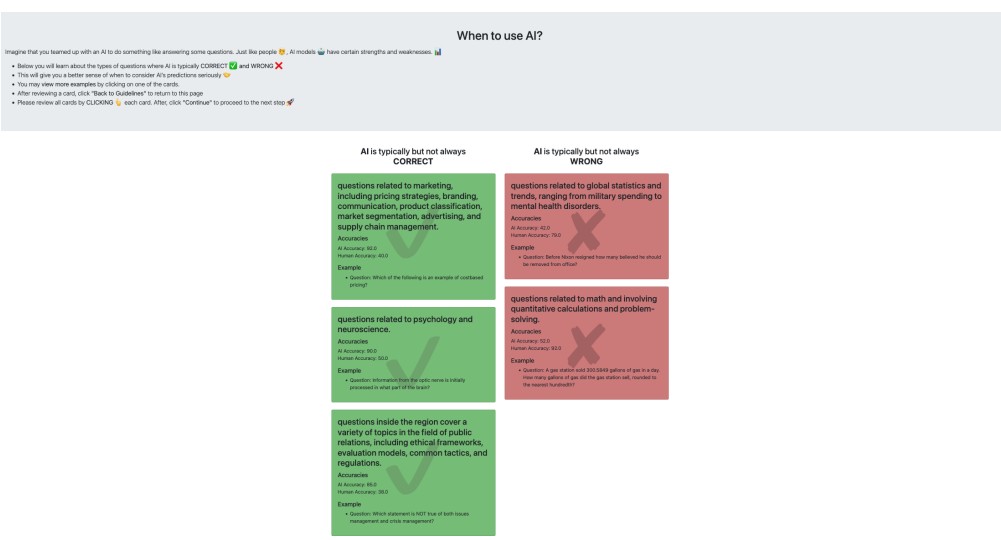

(b) Initial instructions for the teaching phase of the onboarding process.

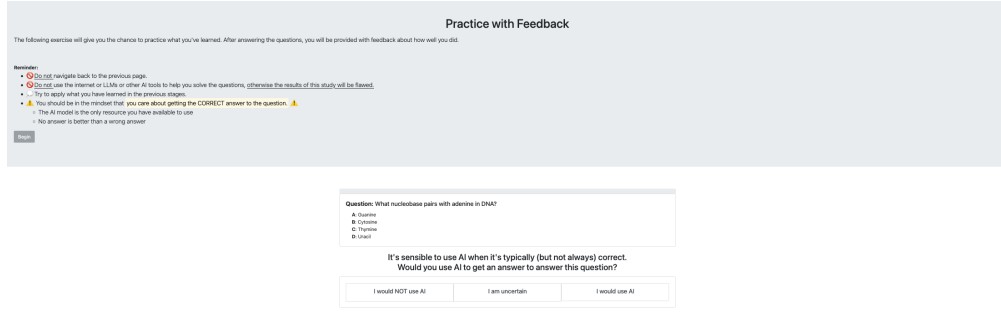

(c) Instructions shown to users before the practice phase.

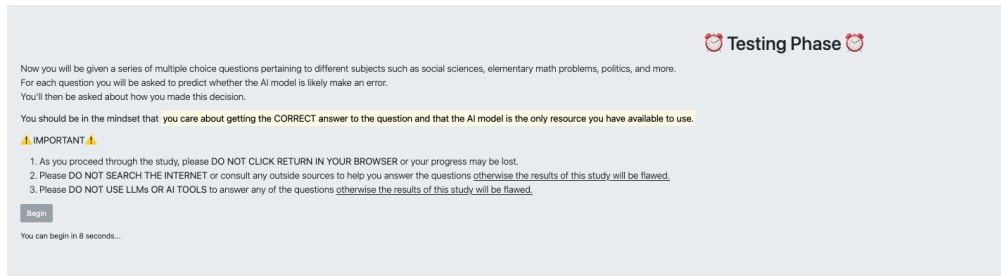

(d) Instructions shown to users before the testing phase.

Figure 7: Screenshots from the user study interface showing the instructions given to users.

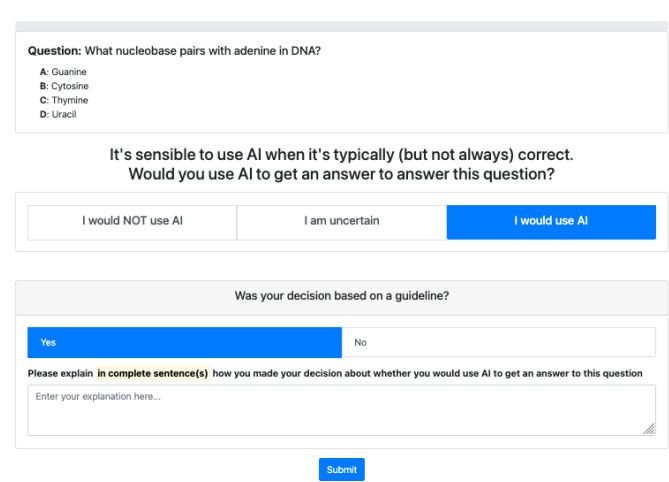

(a) An example question shown to a user during the teaching phase. They are required to answer whether or not they would use AI or not and explain how they came to their decision.

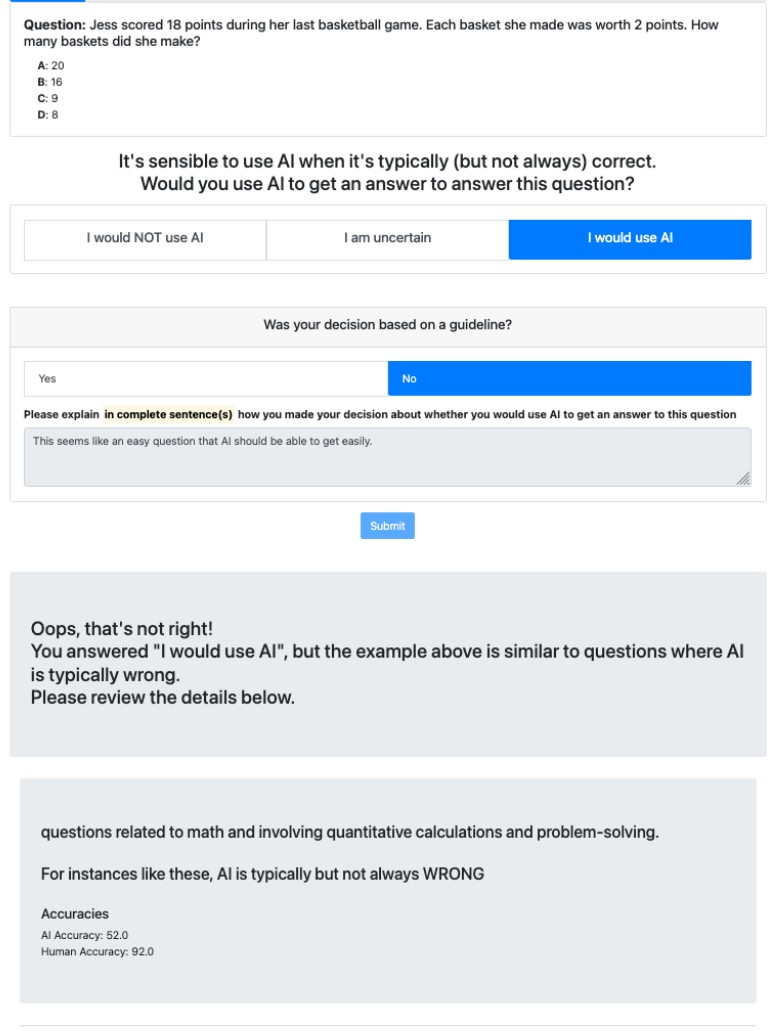

(b) An example of feedback given to a user who answers incorrectly during the teaching phase.

Figure 8: User study screenshots showing the feedback process during onboarding.

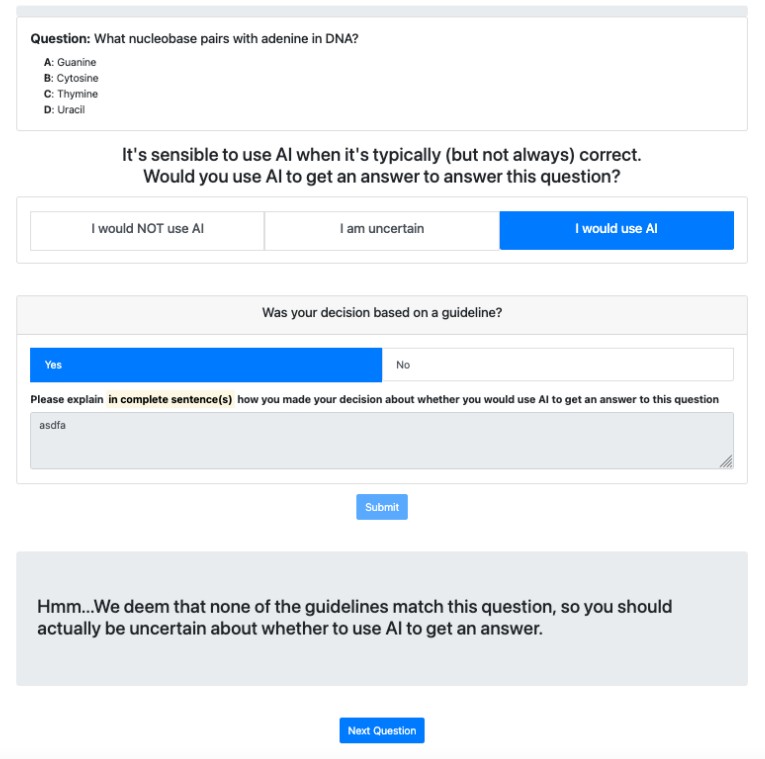

(a) An example of feedback given to a user who answers uncertain during the teaching phase. Since in practice, users will not always encounter instances which match any of the guidelines they have learned, we provide them this option.

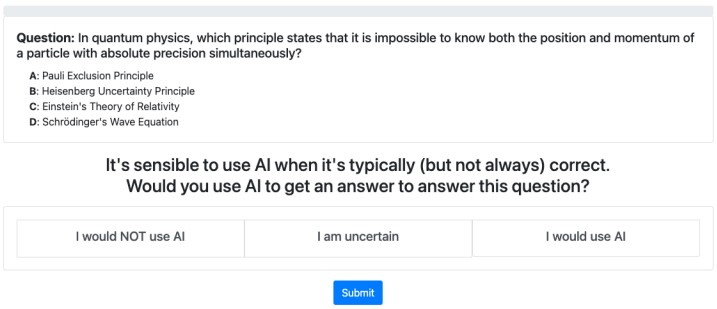

(b) An example question shown to a user during the testing phase. No feedback is given at in this stage and users are evaluated based on whether they select the correct options based on the onboarding.

Figure 9: User study screenshots continued from 8

| Standard | Description |
|---|---|
| `4.MD.A.2-fraction` | Use the four operations to solve word problems involving distances, intervals of time, liquid volumes, masses of objects, and money, including problems involving simple fractions or decimals, and problems that require expressing measurements given in a larger unit in terms of a smaller unit. Represent measurement quantities using diagrams such as number line diagrams that feature a measurement scale. |
| `8.EE.C.7` | Solve linear equations in one variable. |
| `5.OA.A.1` | Use parentheses, brackets, or braces in numerical expressions, and evaluate expressions with these symbols |

Table 3: Example of standards from MathCAMPS that with varying levels of description-complexity. This could affect the ease with which error description methods can generate them.

| Dataset | Example |
|---|---|
| MMLU-Math | Let $p = (1, 2, 5, 4)(2, 3)$ in $S_5$. Find the index of $\langle p \rangle$ in $S_5$. |
| MMLU-Health | Which of the following structures is derived from ectomesenchyme?
A: Motor neurons    B: Skeletal muscles    C: Melanocytes    D: Sweat glands |
| MathCAMPS | A truck driver starts his route with 9 gallons of gas in his tank. He stops at a station and adds to this tank another 21/36 gallons of gas. Later, he stops at another station and adds another 26/42 gallons of gas. How many gallons of gas total does the truck driver have now in his tank? |

Table 4: Example questions from MathCAMPS and MMLU.

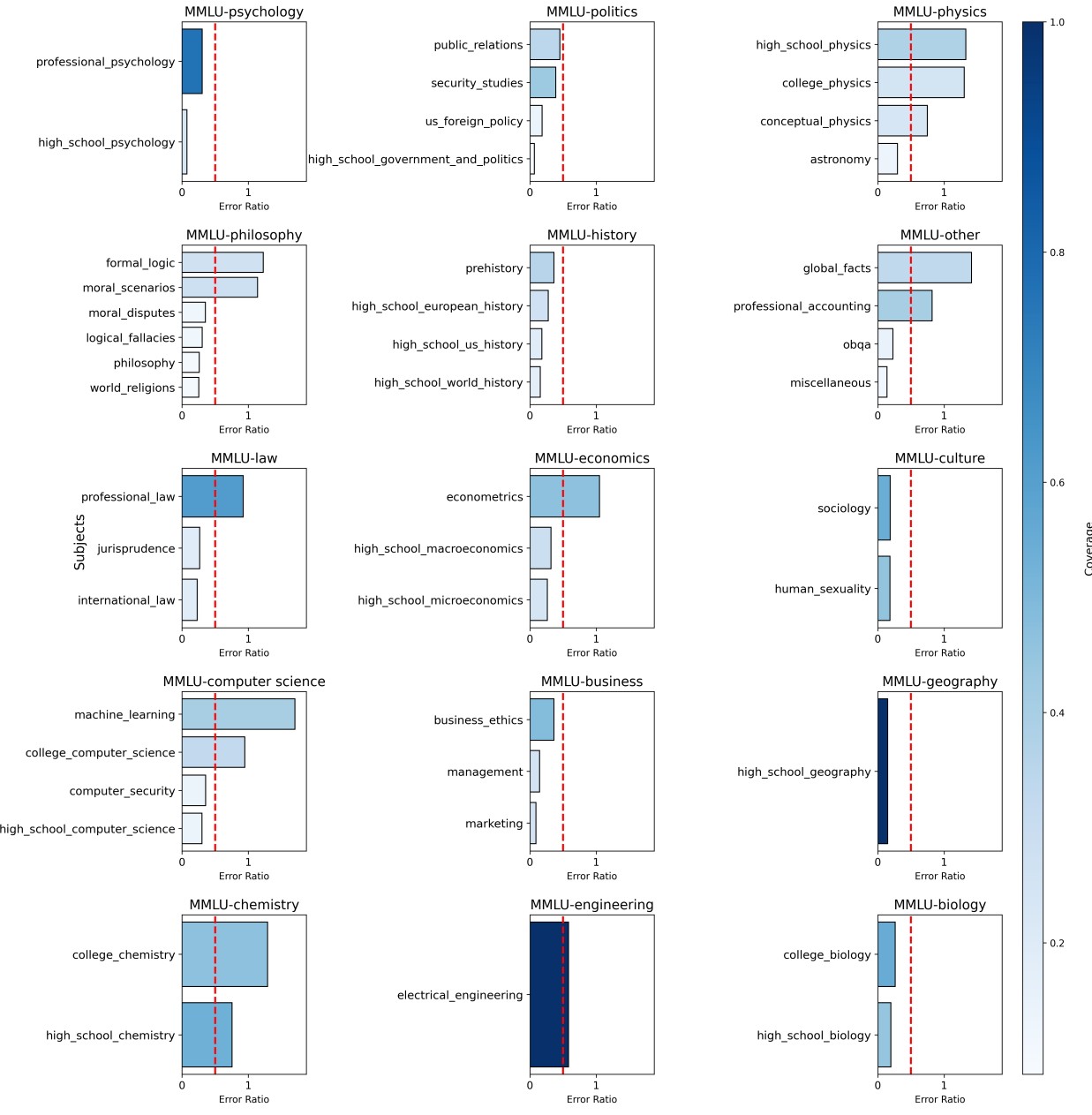

Figure 10: Subjects from other subcategories of **MMLU** sorted by error ratio for `GPT-3.5-turbo`.

