# OpenReview forum: "Teaching People LLM’s Errors and Getting it Right"
_TMLR — Rejected by TMLR_

### Review · Reviewer_iTa5 · 2026-03-15

**Summary Of Contributions:**

Summary:

The paper investigates whether it's possible to teach people which tasks an LLM can be trusted on. Specifically, the authors examine why prior work [1] found no positive impact on human-AI team accuracy from such teaching. They analyze this negative result along three axes: (1) they rule out the absence of common failure patterns, showing that meta-label groups with high coverage and error ratio do exist (Section 4); (2) they find that current automated methods for discovering and describing these patterns yield mixed results; simple prompting works reasonably well, but the embedding-based method used in [1] does not, suggesting this stage is the key bottleneck (Section 5); and (3) they demonstrate sensitivity to the choice of evaluation metric; showing that replacing team accuracy with a measure of whether users can predict when the LLM will fail reveals a significant positive teaching effect (Section 6).

Strengths:
* S1: The paper is very systematic. It thoroughly investigates why prior work [1] found negative results and investigates potential causes. Such analytic research is unfortunately often undervalued in the broader ML literature, but the analysis is overall well-designed in this work.

* S2: The metrics (coverage, error rate, “can I trust my LLM” measure) are well-motivated. Regarding the “can I trust my LLM”: this metric distinguishes between teaching effectiveness (i.e., can users predict when the LLM will fail?) and the user’s own task ability (i.e., can they answer correctly themselves?). By contrast, team accuracy conflates the two.

* S3: The analysis on automatically finding failure patterns in Section 5 is very thorough and tests a lot of variations (choice of LLM, with or without CoT).

* S4: The paper openly discusses limitations. This transparency is appreciated.

Weaknesses:
* W1: The user study is narrow and biased. Only 20 participants were recruited, all from the authors' personal network (NLP PhD students and friends). This raises several concerns: (a) participants may have been (inadvertently) aware of the study's intent. The authors even note they chose this recruitment strategy to "more easily communicate the intent of the study." (b) the participants are non-representative, as presumably all participants have higher education and/or technical backgrounds; and (c) the NLP background is a potential confound, since these participants likely had prior exposure to MMLU questions and may already hold informed priors about LLM failure modes. A user study ideally captures a participant's first unprimed interaction with the problem. Any information flow outside the user study protocol (e.g., oral clarifications, shared context with colleagues) must be avoided.

* W2: The study measures the effect of teaching by presenting the same 20 questions before and after the intervention. This makes it difficult to disentangle genuine learning from memory or anchoring effects, as participants may simply recall their earlier responses or the questions themselves.

* W3: Only 20 user study questions with failure patterns and 5 without are used. This limits the statistical power of the finer-grained analyses (e.g., Figure 4 right).

* W4: The “do i trust my LLM” metric, while well-motivated (cf. S2), has its own gap. It measures whether users can predict LLM failures, but not whether this prediction ability actually translates into improved decision-making or outcomes. A user might learn to correctly flag unreliable LLM answers yet still lack a viable alternative;  in which case the practical benefit of teaching remains unclear.

* W5: Both datasets involve well-defined questions with unambiguous correct answers. It is unclear whether the findings generalize to tasks where correctness is harder to define, such as open-ended generation or summarization, or to domains where failure patterns may be qualitatively different. Coding would be a particularly relevant case given the rise of tools like Claude Code or Codex, where engineers delegate tasks to LLMs with sometimes limited oversight.
* W7: The user study reuses the failure patterns generated by Mozannar et al. [1]; the very method the authors previously identified as the weak link. This is an odd choice and a user study with better found failure patterns would be a valuable addition. For example, does team accuracy benefitin this case or still not?

* W8 (likely out-of-scope for the present work): Across model revision failure patterns might be unstable. I.e., something that worked previously fails in the new model revision or something that failed previously works now. This’d be an important question to see the long-term value of teaching effectiveness. However, this is likely somewhat out-of-scope for the present paper.

---

[1] https://arxiv.org/abs/2311.01007

**Additional Comments:**

Comments:
* C1: Consider citing [1] in the abstract, as the paper is specifically an analysis of that work rather than of prior work in general.
* C2: Figure 3 would benefit from human-readable labels instead of the cryptic MathCAMPS codes.
* C3: The table in Figure 4 (right) should be moved within the page limits.

Questions:
* Q1: Could you measure whether improvement in failure prediction correlates with better decision-making under better failure patterns? This relates to W4 & W7: the current metric shows users learn to predict failures, but the practical downstream impact remains undemonstrated.
* Q2: How is the function "has_pattern" operationalized?

**Audience:**

Yes

**Audience Explanation:**

This is a very relevant topic and, thus, there's an audience for this work.

**Claims And Evidence:**

No

**Claims Explanation:**

The claims are partially supported:

* The claims regarding the existence of failure patterns (Section 4) and the bottleneck in automated discovery methods (Section 5) are well-evidenced through systematic evaluation across multiple LLMs and datasets (S1, S3).
* However, the central claim that teaching is effective rests on a user study with a small, non-representative, and potentially biased participant pool (W1), confounded by reuse of the same questions pre/post-teaching (W2), and using the very failure patterns discovered by the method that the paper identifies as the weak link (W7).

**Requested Changes:**

* The user study must be strengthened: larger, representative participant pool (W1), distinct question sets pre/post-teaching (W2), and ideally testing with higher-quality failure patterns such as those from the Direct method (W7).
* An additional data setting would be very appreciated (W5).

---

### Review · Reviewer_fz67 · 2026-03-30

**Summary Of Contributions:**

This paper investigates why prior work on the AI-integration teaching pipeline fails to reduce human overreliance on LLMs. The authors systematically analyze three stages of the pipeline: (1) existence of failure patterns, (2) automatic discovery of such patterns, and (3) evaluation metrics for teaching effectiveness. They show that meaningful failure patterns do exist, but current automated methods (especially embedding-based ones) struggle to recover them accurately. Finally, they propose an alternative evaluation metric, users’ ability to predict LLM failures—and demonstrate via a user study that teaching improves this ability, even when traditional human-AI team accuracy does not reflect gains

**Audience:**

Yes

**Audience Explanation:**

This paper investigates why prior work on the AI-integration teaching pipeline fails to reduce human overreliance on LLMs. The topic is interesting for people doing applied AI research.

**Claims And Evidence:**

No

**Claims Explanation:**

The paper directly evaluates each research question (RQ1–RQ3) with targeted experiments. For example, it shows that failure patterns do exist using quantitative measures (coverage and error ratio) across datasets like MMLU and MathCAMPS. This provides clear evidence supporting the claim that Stage 1 is not the bottleneck. However, experiments rely heavily on benchmark datasets with meta-labels. It is unclear whether the same conclusions hold in real-world settings where such labels are unavailable. The user study was also done on a small sample size (only 20 participants).

**Requested Changes:**

1. There should be a discussion on how robust are the findings across different prompting styles or more recent LLMs.
2. Evaluate on more realistic, unlabeled datasets where failure patterns must be discovered without meta-labels. Failure patterns must be discovered automatically from raw data.
3. Expand the user study with more diverse participants and larger sample size.
4. Include an ablation study showing how each stage contributes to overall performance.

---

### Review · Reviewer_wVo9 · 2026-04-25

**Summary Of Contributions:**

The paper revisits a negative result from Mozannar at al. 2023 that showed that onboarding (e.g., teaching humans about AI errors) did not have a significant positive effect on human-AI collaboration task accuracy on MMLU. They for examine several hypotheses as to why the result may be negative, including 1. lack of common error patterns, 2. whether methods can identify and cluster them, and 3. test accuracy may be too high of a bar to show the benefits of onboarding on a benchmark such as MMLU where humans have a lower base accuracy, as humans may be able to better identify when to trust an LLM or not even if they still cannot identify the correct answer in the latter case. The analysis in the paper shows that 1. is not the explanation but 2. and 3. may have attributed to the negative result in Mozannar at al 2023, due to their choice of clustering methodology (for 2.) and the fact that on a newly conducted user study, users did improve on their ability to identify when to trust an LLM (for 3.).

**Audience:**

Yes

**Audience Explanation:**

See above.

**Claims And Evidence:**

Yes

**Claims Explanation:**

The paper addresses an important and timely question and provides concrete insights for future investigation into and efforts around human AI collaboration (e.g., the importance of the methodology used to identify failures -- prompting works quite well whereas the previously used in embedding method may not; task accuracy alone may not capture the benefits of teaching humans). That said, the newly proposed metric coupled with the study design feels almost tautologically true -- by telling users about the mistakes that a given model makes on questions in a given benchmark, we would expect that users would be able to perform better in choosing which LLM answers are mistakes or not (e.g., as if giving them high-level answers to the test). Moreover, given the use of the same dataset-model pair throughout the pipeline (i.e., summarize the mistakes for this dataset-model pair for user to better assess when to trust a model for that dataset), it's not clear how to generalize this pipeline in practice, including when there are not already labeled failures for a given dataset-model pair.

Also, as a some comment, it might be more intuitive to use error-rate (i.e., errors over total) instead of error-ratio (i.e., errors over non-errors). Also, parts of the appendix were especially helpful for making the study concrete and may help if moved to the main paper in favor of the textual descriptions (e.g., example group descriptions in Appendix C), or at least referenced more often directly in the main paper.

**Requested Changes:**

See above.

---

### Comment · Reviewer_wVo9 · 2026-04-12
**Submission Review**

The paper revisits a negative result from Mozannar at al. 2023 that showed that onboarding (e.g., teaching humans about AI errors) did not have a significant positive effect on human-AI collaboration task accuracy on MMLU. They for examine several hypotheses as to why the result may be negative, including 1. lack of common error patterns, 2. whether methods can identify and cluster them, and 3. test accuracy may be too high of a bar to show the benefits of onboarding on a benchmark such as MMLU where humans have a lower base accuracy, as humans may be able to better identify when to trust an LLM or not even if they still cannot identify the correct answer in the latter case. The analysis in the paper shows that 1. is not the explanation but 2. and 3. may have attributed to the negative result in Mozannar at al 2023, due to their choice of clustering methodology (for 2.) and the fact that on a newly conducted user study, users did improve on their ability to identify when to trust an LLM (for 3.).

The paper addresses an important and timely question and provides concrete insights for future investigation into and efforts around human AI collaboration (e.g., the importance of the methodology used to identify failures -- prompting works quite well whereas the previously used in embedding method may not; task accuracy alone may not capture the benefits of teaching humans). That said, the newly proposed metric coupled with the study design feels almost tautologically true -- by telling users about the mistakes that a given model makes on questions in a given benchmark, we would expect that users would be able to perform better in choosing which LLM answers are mistakes or not (e.g., as if giving them high-level answers to the test). Moreover, given the use of the same dataset-model pair throughout the pipeline (i.e., summarize the mistakes for this dataset-model pair for user to better assess when to trust a model for that dataset), it's not clear how to generalize this pipeline in practice, including when there are not already labeled failures for a given dataset-model pair.

Also, as a some comment, it might be more intuitive to use error-rate (i.e., errors over total) instead of error-ratio (i.e., errors over non-errors). Also, parts of the appendix were especially helpful for making the study concrete and may help if moved to the main paper in favor of the textual descriptions (e.g., example group descriptions in Appendix C), or at least referenced more often directly in the main paper.

---

### Decision · Action_Editor_WzAn · 2026-06-02

**Recommendation:** Reject

**Additional Comments:**

Given that the authors are unable to participate in the discussion, a major revision is expected to address all concrete concerns raised by the reviewers.

**Audience:**

Yes

**Audience Explanation:**

The paper investigates whether people can be taught to determine which tasks they can rely on LLMs. Reviewers find this research question relevant and well-motivated, with implications to understanding and improving human-AI collaboration at large.

**Claims And Evidence:**

No

**Claims Explanation:**

Multiple reviewers have raised questions about the user study; specific concerns include the validity of observations from a small biased sample size (20 participants recruited through personal networks), and lack of analysis/discussion that addresses possible confounding factors in the experiment design. Reviewers also raised questions about generalizing the results to tasks without labeled failure patterns, and the design choice to rely on the patterns from [Mozannar et al. 2023].

**Resubmission Of Major Revision:**

The authors may consider submitting a major revision at a later time.